# On James Sterba's Refutation of Theistic Arguments to Justify Suffering

Bruce R. Reichenbach

Department of Philosophy, Augsburg University, Minneapolis, MN 55454, USA; reichen@augsburg.edu

**Abstract:** In his recent book *Is a Good God Logically Possible*? and article by the same name, James Sterba argued that the existence of significant and horrendous evils, both moral and natural, is incompatible with the existence of God. He advances the discussion by invoking three moral requirements and by creating an analogy with how the just state would address such evils, while protecting significant freedoms and rights to which all are entitled. I respond that his argument has important ambiguities and that consistent application of his moral principles will require that God remove all moral and natural evils. This would deleteriously restrict not only human moral decision making, but also the knowledge necessary to make moral judgments. He replies to this critique by appealing to the possibility of limited divine intervention, to which I rejoin with reasons why his middle ground is not viable.

**Keywords:** problem of evil; James Sterba; existence of God; theodicies; moral evil; natural evil; ethical principles

James Sterba seeks to reinvigorate the argument formalized by John Mackie against God's existence from the presence of evil (Mackie 1955). Although his immediate target is the Free-Will Defense argument advanced by Alvin Plantinga (1974), as he proceeds through his book, he critiques theodicies advanced by other theists, all in support of his contention that no extant greater good defenses or theodicies successfully show that the degree and amount of evil that exists in our world is compatible with God's existence (Sterba 2019, p. 11). While formulating his argument deductively, including as a logical *reductio* (Sterba 2019, pp. 189–90), he contends at the same time that "the problem of evil is fundamentally an ethical, not a logical or epistemological, problem" (Sterba 2019, p. 5). The reason is that "a defensible solution depends on a moral requirement that applies to both God and ourselves and to the logical relations of that principle to the circumstances in which we find ourselves" (Sterba 2019, p. 32 n18). He sees his unique contribution in stressing the ethical structure underpinning the discussion and the fact that if a good God exists, he has not satisfied those requirements.

After advancing Sterba's version(s) of the atheologian's arguments from *moral* evil, I will attempt to clarify the terminology used, since the discussion in important ways trades on it. Following that, I will develop my critique of Sterba's arguments and engage his responses to my critique. In the final sections I will consider his discussion of *natural* evil and the principles and requirements he invokes with regard to God's obligation to prevent it. I do not pretend to claim that my responses in defense of a greater good theodicy are unique or novel. Indeed, as I will point out in his replies, he has often anticipated many of them. However, I will focus on his ethical principles and argue that they and the arguments they generate are inadequate and that theists can reasonably defend their position.

## 1. Sterba's Arguments from Moral Evil

The initial question that Sterba poses is "Why then, in the actual world, couldn't God . . . be more involved in preventing evils that result in the loss of significant freedom for their victims?" (Sterba 2019, p. 20). Or more generally, "Why does not God prevent significant suffering or loss when he is morally obligated to do so and could do so easily?"

His response, and the burden of his book, is that greater good defenses, invoking freedom, soul-building (Adams 1999), and skeptical theism (Bergmann 2009), offer inadequate answers (Sterba 2020a, p. 203). He provides a summary of his main argument regarding moral evil to parallel that proposed by Mackie.

1. "There is an all good, all powerful God."
2. "If there is an all good, all powerful God then necessarily he would be adhering to Moral Evil Prevention Requirements I–III."
3. "If God were adhering to Moral Evil Prevention Requirements I–III, then necessarily significant and especially horrendous evil consequences of immoral actions would not be obtaining through what would have to be his permission."
4. "Significant and especially horrendous evil consequences of immoral actions do obtain all around us, which, if God exists, would have to be through his permission.
5. "Therefore, it is not the case that there is an all good, all powerful God" (Sterba 2020a, p. 208).

His Moral Evil Prevention Requirements, mentioned in premise 2, are

I. "Prevent, rather than permit, significant and especially horrendous evil consequences of immoral actions without violating anyone's rights (a good to which we have a right) when that can easily be done. For example, if you can easily prevent a small child from going hungry . . . without violating anyone's rights then you should do so" (Sterba 2019, p. 126).

II. "Do not permit, rather than prevent, significant and especially horrendous evil consequences of immoral actions simply to provide other rational beings with goods they would morally prefer not to have" (Sterba 2019, p. 128). For example, do not allow someone to be a suffering victim so that another person can aid them and relieve them of their victim-sufferings.

III. "Do not permit, rather than prevent, significant and especially horrendous evil consequences of immoral actions (which would violate someone's rights) in order to provide such goods when there are countless morally unobjectionable ways of providing those goods (Sterba 2019, p. 128; 2020a, pp. 204–6)." That is, when you can, do not let them be a victim in the first place.[1]

Premise 1 is assumed for the reductio argument, while 5 follows validly. Premise 4 is true, although Sterba's defense of it is problematic and leads to our major objection, as we shall see shortly. Premise 2 reflects Sterba's recrafting of Mackie's argument by appealing to ethical considerations. We will especially focus on premise 3, which is the central, hypothetical claim about what would happen if premise 2 were implemented.

Although Sterba does not overtly formulate the following moral principle, his discussion presupposes it: (MP) A good being, like a just state, will perform all moral actions in its power that will prevent significant or horrendous evil and/or promote significant freedoms and rights when it can be done easily, without the net loss of significant freedom and rights to which all are entitled, even when doing so requires interfering with the freedoms and rights of some.

We can use a case study advanced by Sterba to illustrate MP. Both rich and poor have a right to resources to satisfy their basic needs. Lacking such resources is an evil. Consider a situation where the rich have more than enough resources to satisfy their basic needs, whereas the poor lack those resources, although they have tried to acquire them

---

[1]　Sterba suggests an alternative formulation of his argument in more positive terms of providing goods for which we have a right, rather than preventing loss of rights and freedoms. The state has an obligation to provide for its citizens those goods to which they have a right, when it can easily do so, so long as it does it in a way that does not violate the morally significant rights of others. The rights of others may be violated only if the exercise of those rights involves serious wrongdoing. Because of God's power and knowledge, if God existed, God would be able to provide for us, God's citizens, without violating morally significant rights and as morally good is obligated to do so. God is not logically constrained from doing this, otherwise God would be weaker than humans are. However, it is apparent that God has failed in this duty. Therefore, God does not exist (Sterba 2020a). Sterba sees this formulation of his argument as equivalent to the above argument, since "the nonprovision of goods to which we have a right is a way of doing evil" (Sterba 2020a, p. 204).

legitimately. In such a situation, the poor have a right to take the surplus resources from the rich and the state has the obligation not to interfere with their doing so. The rights of the rich to their excess resources are not denied, since they have earned them, but the freedom to meet one's basic needs takes precedence over the freedom to use justly earned but non-basic or excess goods (Sterba 2019, pp. 15–17).

## 2. Argument from the Pauline Principle

Sterba suggests a second formulation of his argument, this time from what he terms the Pauline Principle.

6.  Pauline Principle: One should not do or allow evil so that good will come of it (Sterba 2019, p. 2).
7.  According to the traditional free will defense, God allows moral evils so that the goods of freedom of choice and freedom of action are possible. Similarly, in the soul-building theodicy, God allows evils so that the good of character development is possible.
8.  Therefore, the traditional free will defense and soul-building theodicy are incapable of justifying moral evil.

Sterba recognizes that there are exceptions to the Pauline Principle. These exceptions have to do with trivial offenses, reparable offenses, or avoiding serious or far greater harm to innocents.

9.  Exceptions to the Pauline principle are allowed "when the evil is trivial, easily reparable, or the only way to prevent a far greater harm to innocents" (Sterba 2019, p. 50).
10. These exceptions arise because humans lack the power to arrive at the good or avoid or prevent the evils (Sterba 2019, p. 50).
11. God is omnipotent and omniscient.
12. Therefore, God could avoid these exceptions by using his power and knowledge. For example, he can act earlier in the causal chain (Sterba 2019, p. 50). Put another way, God always has the causal powers "to prevent the greater evil without permitting the lesser evil" (Sterba 2019, p. 57), and there is no logical contradiction in exercising that power (Sterba 2020a, p. 205).
13. However, God has not avoided these exceptions. See premise 4 above.
14. Consequently, "none of these exceptions to the Pauline Principle that are permitted to agents, like us, because of our limited power, would hold for God" (Sterba 2019, p. 50).

This is a stronger conclusion than usually given by Sterba, who qualifies conclusion 14 when he analogizes God to the just political state.

> *God, like a just political state, should not try to prevent every moral evil. Instead, like a just political state, God should focus on preventing the significant moral evils that impact people's lives. God should not seek to prevent lesser evils because any general attempt to prevent such evils would tend to interfere with people's significant freedoms. (Sterba 2019, p. 59)*

We would not want, he affirms, a political or divine police state where to remove all lesser evils all freedoms would be curtailed. Thus, God, like the state, should concentrate on significant evils.[2]

However, given God's omni-properties and the fact that God can intervene anywhere along the causal chain while protecting significant freedoms to intend evil, and given the ambiguity and relativity of "significant" (which we will argue in the next section), the stronger conclusion 14 follows. In a Sterba-type argument, God would have no excuse

---

2   Although Sterba does not go this direction, building on the analogy between just states and God, he might argue that the presence of unjust states also constitutes an argument against God's existence, for if God existed, he would be able to and should prevent the existence of unjust states that promote moral evil, remove significant freedom, and disregard rights to which all are entitled.

for permitting both significant and lesser moral evils, given his omni-properties and the Pauline Principle. In short, specifically referencing the freedom defense, "if God is to be justified in permitting such moral evils, it has to be on grounds other than freedom because an assessment of the freedoms that are at stake require God to act preventively to secure a morally defensible distribution of freedom" (Sterba 2019, pp. 23–24).

### 3. Setting the Stage

Moral evils may be defined as instances of pain, suffering, loss, dysfunction, and states of affairs significantly disadvantageous to living beings that are caused by actions for which human agents can be held morally blameworthy. Natural evils are instances of pain, suffering, loss, dysfunction, and states of affairs significantly disadvantageous to living beings that are caused by actions for which humans cannot be held morally blameworthy.[3] This classification does not differentiate between moral and natural evils based on the types of results, but rather with reference to the moral accountability of the agents or causes.

The question Sterba raises for the theist is why a just, omnipotent, and omniscient God permits significant moral and natural evils, when presumably God could easily prevent them by altering the causal conditions somewhere along the causal chain. For Sterba, the contradiction between unjustifiable, significant existing evils and an all good, omnipotent, and omniscient God, given that Sterba's moral requirements apply to God as well as to us, provides good reason to think that God does not exist.

Sterba is not interested in ordinary or less significant moral and natural evils, but wants to focus on significant and horrendous evils. He notes Marilyn Adams' definition of horrendous evils, but primarily directs his attention to "significant moral evils that have their origin in human freedom and the lack thereof" (Sterba 2019, p. 14). He characterizes "significant moral evils" as the significant negative consequences of our immoral acts (Sterba 2019, pp. 12, 23, 26, 28). In his book, Sterba frames much of his discussion of significant evil in terms of freedoms that are lost. The freedom he has in mind is not the freedom necessary for making moral decisions, but freedoms to which we have a right that a just society would preserve or defend. Such freedoms include freedom from assault (Sterba 2019, p. 13), from lacking resources to "satisfy basic needs" (Sterba 2019, p. 5), from disproportionate distribution of goods and resources (Sterba 2019, p. 18), from "unjust economic systems" (Sterba 2019, p. 20), and from being unable to live out our life without being tortured or killed (Sterba 2019, p. 20).

Sterba notes his differences from Plantinga and other free will theists regarding the freedom invoked in their defenses/theodicies. Whereas Plantinga appeals to the freedoms necessary for making morally significant choices, Sterba wants to narrow the freedoms to those that "a just political state would want to protect since that would fairly secure each person's fundamental interests" (Sterba 2019, p. 12). Sterba holds that God would have more reason to defend interests in his sense of social freedom than in Plantinga's sense of choice-making freedom.

In speaking of significant evils and significant suffering, Sterba holds that evil can be qualified and quantified; there are "degrees" and "amounts" of evil in the world (Sterba 2019, p. 1). However, it is also important to note that what constitutes significant and insignificant, acceptable and unacceptable, suffering is relative to persons, contexts, and even outcomes (for example, whether suffering is the final outcome or whether suffering is a means to a greater good or a byproduct of some action). Some people tolerate pain and suffering more readily than others. Some children are more pain intolerant than adults; the bodybuilder more accepting than the couch-potato. Some people take the loss of a partner or relative much harder than others. Whereas defamation or election loss is a significant

---

3　Sterba does not define moral and natural evils, with the result that his distinction is unclear. For example, he treats climate change as a natural evil, while accepting that humans are at least partly responsible for it. Sterba here appears to be using "responsible" in a moral sense (Sterba 2019, p. 31). Again, he terms a parent giving permission for a child to have surgery to save her life as natural evil (Sterba 2019, p. 98). However, this is not a case of natural evil, for the parent, in intending a good outcome or obeying a rule of beneficence, is morally praiseworthy. The reason for the surgery, however, might involve natural evils.

evil for some, physical attack would be a worse evil for others. In effect, the amount and kind of suffering that might be insignificant to one person will be significant to another, and vice versa. Not only is what constitutes significant suffering relative to persons, it is also relative to other suffering. We measure instances of suffering against each other. For example, physicians ask patients to report the severity of their pain on a scale of 1 to 10. Thus, in a world where we normally experience pains at level 3, level 8 pains will be very significant. In another world where we normally experience no pains, level 1 pains may be very significant, if not horrendous.

The matter of significance, whether of significant evil or of significant freedom, becomes further muddied when Sterba contrasts "lesser freedoms" with "more significant freedoms" (Sterba 2019, p. 29) and "lesser evils" with "significant evils" (Sterba 2019, p. 51). On the one hand, Sterba might be understood to hold that lesser freedoms and lesser evils are *in*significant. However, the relativity of determining significance on this understanding is precisely the point made above. On the other hand, if "lesser" still leaves the freedom and evils to be significant, his attempt to have God focus on significant evils leaves no contrast, leading to the contention that God should remove all evils. We will return to this important point when we inquire whether Sterba's position requires that God meticulously remove all instances of suffering or loss.

### 4. Sterba's Defense of Premise 2

Sterba contends in premise 2 that if there is an all good, all powerful God, then necessarily he would adhere to Moral Evil Prevention Requirements I–III. To motivate this, Sterba creates an analogy between the just state and God. Within a state, significant freedoms, which are freedoms in terms of rights that every human has or deserves, "are those freedoms a just political state would want to protect since that would fairly secure each person's fundamental interests" (Sterba 2019, p. 12). Political states are obligated to secure these freedoms by law, "even when doing so requires interfering with the freedoms of some of their members" (Sterba 2019, pp. 12–13). This interference can be justified only if it is done to protect the freedom of others to which they have a right and "that everyone should have" (Sterba 2019, p. 13). If we fail to interfere, we have a "morally unacceptable distribution of freedom" (Sterba 2019, p. 13).

As good or just, God is morally obligated to follow the same Moral Evil Prevention Requirements as just political states. This includes securing a range of important freedoms based on universal rights, even when doing so requires interfering with individual freedoms of some. That is, God is morally obligated to prevent a morally unacceptable distribution of freedom.

One could question whether an appropriate analogy can be created between the just state and God. After all, the properties of the former are finite, whereas God's properties express his infinity. However, since the soundness of Sterba's argument does not rest on this analogy, which is more illustrative than argumentative, but on the ethical principles or requirements that purportedly govern both, this article will not take up that question.

### 5. Sterba's Defense of Premise 4

To support premise 4—that God has not decreased significant evils that exist by his permission—Sterba appeals to particular cases of significant or horrendous moral evils. We can, he claims, on a case-by-case basis, reimagine the causal sequences that led to the respective tragedy and create scenarios about how God could intervene each time to restrict the less important freedom of the wrongdoers, prevent the suffering, and protect the significant freedom and rights of the participants being victimized. By judicious intervention, God could prevent the rape of a woman, men setting dogs to attack and kill an innocent child, people kidnapping a child, and the sailing of loaded Portuguese slave ships from a Ghanaian port—illustrations provided by Sterba.

I agree that, on a case-by-case approach, one can always speculate about the many ways God could have intervened to prevent the suffering and loss of freedom victims



experience and to further their rights. In this speculation, God, in his causal manipulation of events, would be like Sterba's superheroes (Superman, Wonder Woman, Spider-Man) who, by their valiant actions, create good outcomes stories (Sterba 2019, pp. 19–20). In their fight against malevolent forces, these benevolent, powerful superheroes guarantee that significant freedoms and morally justifiable, universally deserved rights of the victims are protected, even though to do so the superheroes limit the freedoms of those bent on creating evil or mayhem. Similarly, when miscreants intend evil, God might allow them freedom to plan evil but by specific intervention would prevent them from being able to fully carry out their plans. Through his super-knowledge and powerful action, God would intervene either before or during the event to "secure a more important freedom for the would-be victim" (Sterba 2019, p. 130) and thus bring about a world without significant moral evil, though the freedom to entertain evil intentions is preserved.

However, if one is going to construct a theodicy or an atheodicy, general principles, not particular cases, must be the basis for the justification. Otherwise, we look to God to meticulously operate the world to prevent each individual instance of significant or horrendous suffering or to provide the necessary, desired goods. The world would consist of superhero comic book stories, where God is the actor. Sterba's overall argument supports this contention regarding general principles. For example, Sterba believes that skeptical theism, where no justifying reason is provided, fails, for "there is still the need to justify to the victims what would have to be God's permission of the infliction on them of at least the significant and especially the horrendous evil consequences of the actions of wrongdoers. This arises from the very nature of morality, which only justifies impositions that are reasonably acceptable to all those affected" (Sterba 2019, p. 73). This argument depends not on appealing to the possibility of intervention in specific cases, but to general principles of justification. Indeed, Sterba wants to consider whether there is "*a greater good justification for God's permitting significant and especially horrendous evil consequences of immoral actions*" (Sterba 2020a, p. 203). Thus, although Sterba may be correct in contending that theoretically God could intervene in particular cases, his piecemeal justification for the contention that God should universally do so leads to an unsatisfactory situation where God operates the world by meticulous divine intervention.

## 6. Critique of Premise 3

The above considerations pose important issues, but my main worry arises from MP. Sterba distinguishes significant evils from lesser evils. God, he says, like the just political state, need only address the former. However, as argued above, what is one person's lesser suffering might be another person's significant suffering. Significance is a matter of perspective and degree. Are levels 2 or 4 evils significant but only less so, as over against level 7 evils, to be overlooked even if we have the power to remove them without significant negative consequences? Preventing or stopping even so-called lesser or insignificant suffering should be done if one is able to do so easily without creating greater evils or losing significant freedoms to which we all have a right. I should avoid stepping on my neighbor's foot if I can. What generally hinders us from eliminating many evils, as Sterba notes, is our impotence, ignorance of the causal chain, or lack of opportunity or time.

The point here is that, as we argued above regarding MP, to be totally morally good, God should prevent all evils he can, even if God has to interfere with the freedoms of wrongdoers when those freedoms inhibit rights that belong to all. As we have seen, Sterba contends that a God with omni-properties of power and knowledge is capable of so doing. Then, no matter what the number of evils in the world is, if God existed, God could and should be doing more to reduce them (Sterba 2019, p. 66). If God eliminates the highest evils of level 7, then the question arises of why a good, omniscient, and almighty God is not causally involved in the world to remove evils of level 6, since these are now the most significant or serious evils. Moreover, once God removes evils of level 6, evils of level 5 become most significant, if not horrendous, and one wonders what God is doing

about these evils, and so on. The result of such a scenario is the requirement that an all good, omniscient and almighty God is obligated to eliminate all significant moral evils and provide all significant goods where there is no logical impossibility (Sterba 2019, p. 63). Moreover, since "significant" is a relative and comparative term, such that probably no moral evil would not be significant on some person's valuation, God would be required to remove all moral evil. Furthermore, not only is God *obligated*, for Sterba, God *can* do so, since he has the causal power, knowledge, and time to do so (Sterba 2020a, p. 205). "God would never be subject to such causal constraints, and it would be contradictory to assume that he is subject to logical constraints here" (Sterba 2019, p. 129). To accomplish this will require God to meticulously operate the world by divine intervention, either indirectly or directly (by miracle) in a way that would result in the serious curtailment of both morally significant human freedom and the incentive for humans to act beneficently.

This need for continuous divine meticulous intervention becomes clear in Sterba's treatment of Matthew Shepard. He asks what God should do with respect to someone who is mistreated but then goes on to mistreat others. His response is that the intervener should protect the person and significant freedom of the mistreated person in the first place, but then intervene to prevent that person from creating subsequent significant moral evil (Sterba 2019, p. 22).

If God meticulously operates the world by his actions to bring about the good results or the results he desires, there is no reason for us to act. Given God's omni-properties, God can do a much better job at any task than we can. Ultimately, if God is expected to run the world to thereby eliminate all significant moral evils, there is no incentive for humans to act, since God determines what will or will not be done. Even if we do not act, God will intervene to at least meet all basic needs that he can meet, if not do more. There are no situations for humans to act immorally since God prevents all evil consequences; only good can be accomplished. Consequently, there is no opportunity for moral agents to develop their character or engage in soul-building, since there is no morally significant freedom to choose between doing good and doing evil. (Incidentally, this seems to be a difference between God and the just state; it is not the obligation or prerogative of the latter to be engaged in soul-building.) It would be pointless and fruitless to plan or intend evil if the ability to carry out the plans is rendered impossible. Indeed, this scenario not only has moral implications, it has epistemic implications as well. If God meticulously runs the world by direct or indirect intervention, we lack grounds to know how to act, since divine operations replace natural laws (we will return to this later when we address natural evil).

One might expand this scenario beyond the prevention of significant suffering to procuring the good (using Sterba's alternate argument that failure to provide needed goods is an evil). If the just state can easily provide a service (for example, free garbage removal) or goods (plant trees in personal lots to enhance the city) for its citizens without negative impact on its budget or overriding other required duties, benefits, or rights, then it should do so. Failure would count against its goodness or distributive justice. Of course, the just state has limited resources for creating goods for all its citizens. Similarly, a good and benevolent God ought to provide all goods that cost him nothing, and because he has unlimited resources, he can easily do so, thereby demonstrating the beneficence aspect of his moral character. The result of MP is the unacceptable requirement that God meticulously operate the world to remove all instances of (significant) suffering and loss and provide all (basic) goods.

## 7. Limited Intervention

Sterba notes the objection we just made, in particular about soul—or character—building (Sterba 2019, p. 53), and replies that our criticism fails to take account of another option that avoids the necessity of God meticulously running or managing the world, but still meets Sterba's condition that God ought to be preventing significant evils. Sterba argues that a middle ground exists between God always intervening fully (which removes opportunity for moral development) and God not intervening at all (which he takes as the

Plantingian view of the free will response), and that had God existed, he could have used this middle ground. The middle ground, which he terms "limited intervention" (Sterba 2019, pp. 60, 132) or "constrained intervention" (Sterba 2019, p. 90), is that God "not fully intervenes" (Sterba 2019, p. 133) or intervenes in ways that are "only partially successful" (Sterba 2019, p. 132) to leave us room to take action to build our character. God even might allow us to partially carry out our evil intentions, but would step in once the matter becomes a significant evil. He thinks that our freedom would be protected by God allowing us to have "the freedom to imagine, intend, and even to take initial steps toward carrying out (our) wrongdoing," but everyone would be prevented from *fully* implementing their malevolent plans (Sterba 2019, pp. 161, 55).

Soul-building and moral responsibility are made possible because, with God's limited intervention,

> [w]hen you choose to intervene to prevent significantly evil consequences of wrongdo-ers, you will either be completely successful or your intervention will fall short. When the latter is going to happen, God does something to make the intervention completely successful. Likewise, when you choose not to intervene to prevent significant evil con-sequences, God again intervenes but this time not in a fully successful way. In cases of this sort, there is a residue of evil consequences that the victims still do suffer. This residue is not really a significant evil in its own right, but it is harmful nonetheless, and it is something for which you are primarily responsible. You could have prevented those harmful consequences but you chose not to do so and that makes you responsible for them. Of course, God too could have prevented those harmful consequences from happening even if you had decided not to do what you could to prevent them yourself. It is just that in such cases God would have chosen not to fully intervene and completely prevent all the evil consequences in order to leave you with a constrained opportunity for soul-making. Moreover, I maintain that this is exactly what God would be morally required to do. (Sterba 2019, pp. 132–33)

In this way, Sterba holds that limited intervention provides ground for denying that his argument requires God to meticulously operate the world to prevent significant moral evil.

He provides the example of a child being abducted (Sterba 2019, p. 61). With limited interposition, God could allow the kidnapping to occur, giving the bystander opportunity to intervene and develop character. Should the bystander not take any action, God would stop the kidnapping later and rescue the child (for example, by having a policeperson stop the car for a broken taillight). His second example is of someone on the Ghanaian Slave Coast who can warn people not to be tricked into entering the Portuguese slave ships. Should the bystander not act or be unsuccessful, God will use other resources such as the French navy to return the slave ship to port and release the prisoners (Sterba 2019, pp. 132–33). God is the backup plan in case the bystander takes no action or fails in his evil-preventing endeavors.

However, such limited intervention is not an option for Sterba. God's delay in the action and backup role violates Sterba's own Pauline Principle that one should never allow or do evil so that good can come of it. In this scenario, God allows the evil kidnapping of the child or the abduction of Africans on a Portuguese slave ship to occur, so that bystanders can develop character. This appears to be a case where the end of allowing persons (bystanders) to develop their character justifies the evil-producing means, even where the means are only partially successful.

Sterba would reply that we have forgotten that the Pauline Principle can be overrid-den in cases where the harm done is trivial or easily reparable. Accordingly, he might consider these as exceptional cases of trivial or reparable evil (the child is only "somewhat traumatized, but otherwise unharmed" (Sterba 2019, p. 61)). However, that is hardly the case. The kidnapping of the child causes psychological damage to the child. The little good that the bystander could realize from intervening would not compensate for the trauma caused to the child by delay, and even if it did, the principle still would be violated. The

capture of the slaves and their forcible incarceration on the ship leave them more than "a bit traumatized, but otherwise unharmed" (Sterba 2019, p. 132). If Sterba could visit the Cape Coast slave castles in Ghana, as I have done, and see the conditions under which the captured and chained slaves were held in complete darkness with filth up to their knees before they were pushed through a narrow doorway into the foul hold of the slave ship, he would be less sanguine about suggesting that this is a trivial matter. As Sterba notes, "the experience (of significant evil) will almost always be an alien factor in one's life" (Sterba 2019, p. 58). Rather, if God is to be good, he would intervene in the causal event to prevent the abduction of the child or capture of the slaves in the first place, even if he allowed the villains freedom to conceive of their plans. The sufferings and traumatization of the child and captives might appear trivial to Sterba but not to the child and captives. Again, perspective matters on deciding triviality and significance. God "would never be justified in permitting evil in such cases" where the "intrinsically wrongful actions would significantly conflict with the basic interests of their victims" (Sterba 2019, p. 57). "There are no exceptions to the Pauline Principle in this regard" (Sterba 2019, p. 58).

Here is the dilemma. If the evil consequences are trivial and reparable, equivalent to the pain caused by accidentally stepping on someone's foot in exiting the subway, then the bystander's moral character is not significantly involved, and for good reason, since we don't develop moral character in trivialities. If they are not trivial, the Pauline Principle is violated.

Furthermore, on the one hand, on this view of limited intervention, right-doers would soon learn that if they did not act, they need not worry. Not doing anything is justified in that the person believes that a more effective solution would arise, namely, God's intervention. They would have the well justified belief that God will take the necessary, backup rescue action, given his power and character, and that God can do it better than we can. If I act, the slaves' incarceration is temporary; they will be dispatched on the next slave ship. If God intervenes to eliminate the evil, the solution can be maximally effective.

If bystanders saw that God did not intervene immediately the first time but believed that God eventually always intervenes so that suffering is minimal, they would correctly assume that he would do so at other times. Moreover, even if God did not intervene previously, which cannot happen because it would violate God's goodness and power, this provides no reason to think he will not intervene this time (given their adequate theology of God's omni-properties and that God adheres to the Pauline Principle).

If, on the other hand, wrongdoers (or anyone) knew that God would prevent whatever horrendous or significant evil action they planned, there would be no sense in their planning it. Planning for our action presupposes that we believe that we can carry out what we plan. However, if God always intervenes to prevent implementation or to direct anything that happens to his own purposes, they soon would learn that planning was useless because what occurred was planned and brought about by God, not us.

Limited intervention, when it faces Sterba's Prevention Requirements, fails to avoid requiring God to run the world by direct intervention to achieve the end that Sterba demands of God, namely, preventing significant suffering and loss and protecting rights held by all. It ultimately devolves into divine meticulous operation of the world. Rather, "[It] is far more plausible to see an all-good, all-powerful God as also interacting with us continually over time, always having the option of either interfering or not interfering with our actions, and especially with the consequences of our actions" (Sterba 2019, p. 27), the very thesis counter to Sterba's.

## 8. The Limiting of Freedom Objection

Sterba contends that "God could have decreased the moral evil in the world by justifiably restricting the freedoms of some (for example, wrongdoers) to promote significant freedoms for others (victims)" (Sterba 2019, p. 30). This is possible for God, since "an omniscient and all-powerful God would surely be aware of these causal processes as they

get going to divert them or put a stop to them" (Sterba 2019, p. 28). In advancing this, Sterba anticipates another important objection to his atheological case.

> *Now it might be objected that if God interfered with wrongdoing by preventing rather than permitting their significant or even horrendous evil consequences, God would be limiting the wrongdoer's freedom. This is true, but in each and every case where God would thus be limiting a wrongdoer's freedom by preventing rather than permitting significant and especially horrendous evil consequences of his wrongful action, God would also be securing a more important freedom for the would-be victim. So in terms of freedom, it would be better for God to prevent significant or even horrendous consequences of wrongdoing thereby restricting the wrongdoer's freedom than to permit significant or even horrendous consequences of wrongdoing, thereby restricting the freedom of the victim. So any justification in terms of freedom alone (contrary to the Free-Will Defense) would favor the freedom of the would-be victims over the freedom of the would-be perpetrators of wrongdoing. (Sterba 2019, p. 130)*

However, this reply to the limiting-of-freedom objection also falls prey to the Pauline Principle when Sterba argues that "it would be better." God's restriction of the wrongdoer's freedom, which the wrongdoer might claim to violate a basic good, is justified on the grounds that it is better overall that this is done. However, this is nothing less than claiming that one can do evil (restrict freedom) so that a greater good will result, an infraction of the Pauline Principle.

Sterba might reply that the Pauline Principle is not violated, since restricting freedom to do evil is not an evil but a good. The freedoms the state and God should preserve are significant freedoms, that is, freedoms "that would fairly secure each person's fundamental interests" (Sterba 2019, p. 12). The greater good is not freedom per se, but the freedom to do right and the just distribution of freedom. In this, I think, Sterba is correct, but to successfully accomplish this just distribution of freedom still leaves God with having to meticulously administer the world by divine intervention, for he has to determine in each case what freedoms to protect and which to interfere with.

We conclude that Sterba's argument against God's existence from moral evil fails. It imposes too high a cost by making human moral action undecidable and not exercisable. To allow humans meaningful moral freedom and to provide for character development, God must be a risk taker, allowing human choices and action that result in the possibility of moral evil along with moral good.

## 9. Sterba's Argument Regarding Natural Evils

When Sterba turns to natural evils, he applies much the same reasoning to reconciling the existence of God with natural evils as he does to reconciling the existence of God with moral evils. He contends that in our daily life "when the basic welfare of other humans is at stake, in particular, we think we ought to prevent such natural evils from occurring or at least prevent or mitigate their consequences, especially when we can easily do so without causing greater harm to other humans" (Sterba 2019, pp. 157–58). The same applies to preventing the destruction of the basic welfare of living beings in general, whether sentient or not (Sterba 2019, p. 184). By parallel reasoning, he argues, God too is morally obligated to prevent significant and horrendous natural evils to living beings, whether human, sentient, or non-sentient (Sterba 2019, p. 159). It is evident, he believes, that God is not very proactive in preventing significant natural evils to all three types of beings. God's failure to prevent significant natural evils cannot be justified by an appeal to freedom, for allowing them to happen removes or denigrates rather than maximizes the freedom, basic interests, and welfare of those affected. They are diminished, not enhanced, something the just state would not tolerate. Neither can these natural evils be justified by appeal to soul-building, for again not only does the evil overmatch the human soul-building they allegedly make possible, but human soul-building can occur without significant and horrendous natural evils. All that is needed for soul-building are lesser, insignificant, and temporarily delayed evils. In short, God's permission of the consequences of the causes of

natural evil cannot be justified by appeal to either the greater good of human freedom or human moral development.

How should God be acting with respect to humans and nonhuman nature? Sterba argues that, whereas we compete with other living beings and hence cannot always eliminate significant natural evils or their causes, God does not compete with anything else and thus as good is obligated to take into consideration "the interests of all living beings" (Sterba 2019, p. 160). Using his power, God can and should eliminate significant natural evils by divine intervention. There are cases where God cannot intervene. For example, God should be neutral when the conflict is an either/or conflict between nonhumans (as between the predatory spider and the victim fly). However, where one living being is not significantly disadvantaged, God can and ought to prevent the "significant and especially horrendous natural evils upon (humans) and other living beings" (Sterba 2019, p. 159), especially when he can do so "without causing greater harm to other humans" (Sterba 2019, p. 160). Indeed, consonant with what Sterba argued regarding moral evil, a good God should not use significant or horrendous natural evils to protect human freedom or promote human soul-building at the expense of the basic needs of other living beings where the human needs involved are not basic.[4]

Sterba does suggest exceptions. When there is a conflict between humans and nonhuman living beings, God generally ought to prefer human beings. However, even here he gives qualifications. He introduces a Principle of Disproportionality to govern exceptions that favor nonhuman living beings. "Actions that meet non-basic or luxury needs of humans are prohibited when they aggress against the basic needs of individual animals and plants or even of whole species or ecosystems" (Sterba 2019, p. 158). That is, where human basic needs are not jeopardized, God, like us, ought to favor meeting the basic needs of sentient and non-sentient nonhuman beings over non-basic needs of human beings.

## 10. The Principle of Disproportionality

Sterba's Principle of Disproportionality, however, is unacceptable. For one thing, questions paralleling what constitute significant evils arise here with respect to what constitute basic and non-basic needs. Sterba suggests that although we cannot define "basic" and "non-basic" needs, and although we cannot classify all needs in one or the other category, the distinction is not only clear enough to be functional but necessary in moral, political, and environmental philosophy (Sterba 2020b, p. 506 n15).

Maybe so, but how does this distinction get applied? What non-basic human needs would justify intervening in human affairs to protect the basic needs of individual animals and plants? Are not having dandelions in the lawn or spiders and ants in the house (after all, they serve an important function in nature) basic needs, so that one is justified in killing weeds, spiders, and ants? Is having wood for construction a basic need, or should we replace wood with nonorganic building material and thereby stop the lumbering that kills individual trees? After all, trees are living beings with the basic need of life. Is eating meat or seafood or wearing silk clothes, which requires death of sentient beings, a basic need, or is it immoral to not be a vegetarian or to wear silk?

Sterba considers the case of vegetarianism. He writes that "though a more vegetarian diet seems in order, it is not clear that the interests of farm animals would be well served if all of us became complete vegetarians" (Sterba 2020b, p. 508). One reason he suggests is that people would not continue to raise and feed farm animals. However, what right does that violate? Non-existent farm animals do not have a right to be brought into existence. Further, he suggests that being raised under healthy conditions, killed relatively painlessly, and eaten is beneficial to them. True, it is better for animals to be raised in healthy conditions than being raised on an unhealthy factory farm, but how does being killed and eaten benefit them as individual living beings? Life is a basic need, so that killing farm animals in their youth (calves or lambs) or prime justly deprives them of meeting that

---

4  These injunctions follow from his nine Natural Evil Prevention Requirements (Sterba 2019, pp. 184–85).

basic need. His Natural Evil Prevention Requirement IV—"Prevent, rather than permit, significant and especially horrendous evil consequences of natural evil from being inflicted on nonrational sentient beings, as needed, whenever the welfare of rational beings is not at stake and one can easily do so without causing greater or comparable harm to other nonrational sentient life" (Sterba 2019, p. 184)—requires us not to deprive them of their life when human welfare is not at stake, and the vegetarian contends that eating meat is not necessary for or basic to human diets or human welfare. Indeed, if life is a basic need for animals, then killing them to satisfy our desire for meat fails to meet the Pauline Principle, which lies at the heart of his ethic, and animal slaughter is not a trivial or reparable matter, at least to animals.

It is reasonable to conclude that his Principle of Disproportionality, which combines Natural Evil Prevention Requirements IV and VII,[5] is dubious. No one, even in their best moments, could abide by it, let alone ought to. If it is dubious that human beings or the just state does, can, or ought to live by this Principle or these Requirements, there is no reason to think that they also apply to God.

Sterba proceeds to further justify his position that we should maintain farm animals for consumption on the ground that "many will find it difficult to pass up an arrangement that is morally permissible and mutually beneficial for both humans and farm animals." However, his Natural Evil Prevention Requirements show that the arrangement of growing animals and slaughtering them for food, even humanely, does not benefit them. We certainly would not tolerate such a process of raising humans for others' consumption or use on the grounds that it would benefit them. There are, as his Requirements note, "countless morally unobjectionable ways of providing those goods (not required for their basic welfare) to rational" beings, rather than raising animals to be killed and eaten (Sterba 2019, pp. 184–85).

A similar argument might be raised about lumbering. Sterba might argue that lumbering is beneficial in that it thins the trees and thus makes room for a new forest to grow. However, again, this violates not only the Pauline Principle of doing evil (to individual trees by depriving them of the basic need of life) for a greater good (of forest conservation), but also Natural Evil Prevention Requirement VII, according to which we should not prevent natural evil from being inflicted on non-sentient living beings if our welfare is not at stake.

Sterba seems to modify the concept of basic needs by talking about what we as rational beings "need for a decent life" (Sterba 2019, p. 159) or for our welfare. However, what is a decent life? Does welfare go beyond basic needs and goods? Now the debate might be whether a weed-free lawn, an insect-free basement, a house built of wood, and diet that includes meat and fish contribute to a decent life. Might a decent life include even luxury goods, such as art, or is donating to the homeless to be preferred to paying for a visit to an art museum? Even "luxury goods" is not a helpful deciding category. Many Americans consider automobiles essential to a decent quality of life and not a luxury good at all. Some young Americans are not so sure, since they can navigate the city without them. Certainly, my university students in Liberia consider such transport luxury. For many of them, even having a functioning bicycle is a luxury. What might seem basic to one person might be luxury to another, or luxurious to one person basic to another.

Sterba attempts to answer at least part of our objection with his Principle of Human Defense, which

> *permits defense of nonbasic needs of humans against aggression of nonhumans. So while we cannot legitimately aggress against nonhumans to meet our nonbasic needs, we can legitimately defend our nonbasic needs against the aggression of nonhumans seeking to meet their basic needs.* (Sterba 2020b, p. 506 n17)

---

5　　Requirement VII is the same as Requirement IV, except that it applies to non-sentient living beings.

While this self-defense principle does not resolve the problems posed above, it, like human self-defense principles, allows us to defend against ants, spiders, and, with a stretch of the imagination, dandelions. However, like the human self-defense principle, which only allows incapacitation of the aggressor, it does not justify killing them (acting contrary to their basic need of life), only defending against them and removing their capacity to be aggressors. It still leaves problematic issues with the Principle of Disproportionality where our non-basic needs involve the destruction of sentient and non-sentient beings that are not aggressing on us, but that we are using for our benefit or decent life (for example, silkworms, farm animals, and oysters).

In short, not only is the application of basic and non-basic needs and goods ambiguous, but it is dubious that Sterba's Principle of Disproportionality governing human obligations holds true. As such, it is doubtful that it can be used to identify and qualify God's moral obligations with regard to preventing natural evil among all living beings.

## 11. Sterba on God's Obligations to Nature

Returning to the main argument, Sterba contends that God should be preventing the significant and horrendous consequences of natural evils, something that as omnipotent he can do. It is important to note that Sterba applies this to individuals, not just to species. Thus, he worries about the fawn caught in a forest fire. Given Sterba's Natural Evil Prevention Requirements, a good God would be under obligation to rescue the fawn, which he easily could do by causing a quick, localized cloudburst without causing greater harm (Sterba 2019, p. 162).[6] We all sympathize, Sterba notes, with the pitiable, endangered fawn. But what about beetles, snakes, possums, and others likewise trapped in the forest; their biological need for survival is as basic to them as to the fawn and to us, and though we might not be as naturally sympathetic to them as to the fawn, God could and presumably should preserve them as well from the fire. What about non-sentient forest beings: individual pines, aspens, grasses, mushrooms, ferns, wild roses, fungi, and the like? As living organisms, their life is basic to them and threatened by forest fires. Their loss does not occasion any suffering for them but is the loss of life and opportunity to reproduce (pass on their genes). In effect, the consistent application of the contention that God has an interest in living beings, human and non-human, sentient and non-sentient, and ought to preserve their basic needs without discomforting humans would require God not only to rescue the trapped fawn but all the individual insects, mammals, trees, plants, and fungi as well. In effect, God should not allow forest fires, for they cause horrendous destruction and loss—death—of individual living beings and, if animals and plants have rights, their rights, regardless of their sentience (Sterba 2019, p. 162).

Sterba might reply by qualifying his position. Were we to take the interests of all (nonhuman living beings) into account, "we would be in competition with nonhuman living beings such that our survival and basic well-being requires preferring our own interests to their interests in many cases of conflict" (Sterba 2019, p. 160). Preserving all insects and animals and meeting their basic needs would leave us overrun by critters, much to our discomfort. Preserving all vegetable matter and meeting its basic needs would leave us inundated with plants. Hence, preference is given to human needs and "decent living" over the needs of other living beings.

Sterba applies this requirement to give preferential treatment to God as well. God should prefer helping humans because he seeks a special relationship with us. For the theist, this is true, but what moral principle preferences human survival over that of other organisms? Sterba observes that

> *given that it is virtually definitive of traditional theism that God is open to just such a special relationship with us, which, when combined with what I have called a Godly opportunity for soul-making, could ultimately include friendship with God himself,*

---

6   Why Sterba does not consider God bringing about a sudden rainstorm to quell the forest fire a miracle is puzzling, since it would be a specific, intentional, divine intervention in nature (Sterba 2019, p. 162). We will address miracles or divine intervention below.

*then surely God would be morally required to act to prevent significant and especially horrendous evil consequences of natural evil from being inflicted on us when he could easily do so without causing greater harm to other humans. (Sterba 2019, p. 160)*

Thus, if God existed, theistic reasoning about creation, *imago dei*, and even the rationality needed to have relationships with God would play a role in justifying this principle. Of course, if God does not exist, as Sterba holds, this specific support for preferential treatment evaporates.

He goes on to suggest that "meeting our basic needs over those of other species who do not suffer as intensely as we do is the best way to limit serious suffering in the world" (Sterba 2019, p. 161). This too is a dubious claim, considering how many sentient living beings there are in the world in comparison to us. Humans are not the only creatures that suffer. Even if he treats serious suffering qualitatively, it is not obvious that humans suffer more intensely than animals. Watching a cat hit by a car suffer and slowly die in the middle of a road is an unpleasant experience.

Sterba considers whether one might appeal simply to rationality as intrinsically valuable and thereby justify preferencing human needs and decent living, but as he notes, this is a biased perspective. If lions had a say, they would be biased in favor of lions, appealing to their own distinctive traits of excellence. Sterba advances "A Principle of Human Preservation" that gives preference to humans in meeting their basic needs, even at the expense of basic needs of other sentient and non-sentient beings. He justifies it on utilitarian grounds; if the basic needs are not satisfied, it would "lead to lacks or deficiencies with respect to a standard of a decent life" (Sterba 2020a, p. 505). Of course, lions and cows might derive a comparable preferential Principle of Feline or Bovine Preservation, utilizing the same utilitarian argument. With regard to meeting conflicting basic needs, this leads us back to a "might makes right" ethic that Sterba rejects (Sterba 2020b, p. 504).

To summarize, not only are some of his Natural Evil Prevention Requirements and other principles questionable, but also the impossibility of their reasonable application shows the weakness of his natural evil atheodicy. It is doubtful that, for example, Natural Evil Prevention Requirements IV and VII apply to us, let alone to God.

## 12. Natural Evil and Soul-Building

Finally, paralleling his argument about moral evil, Sterba suggests that God's intervention is not inconsistent with soul-building. God could wait a bit in a situation of significant natural evil to give us a chance to act and develop our moral character before he rectifies the situation by taking his own action. In such cases, we would see how God has given us the opportunity to soul-build in the past and can take advantage of that opportunity now, so that we do not become unworthy of heaven (Sterba 2019, p. 95). Consider Rowe's forest fire and the fawn. Where we cannot do anything or fail, Sterba expects God to intervene to save the fawn. However, where we can intervene, God would wait to give us a chance before stepping in and rescuing the fawn (who might suffer a bit in the meantime).

However, the example is fraught with difficulties. For one thing, are we really in a position to see God step in to rescue people and animals in cases of natural evil? What would we be seeing just in case God did (or did not) step in? How would we know it was God who put the fire out rather than it being serendipity? How would we know that God waited for someone to act before he acted, or even if he is waiting for *me* to act? Second, why should we act when we know that not only will God intervene, but that God always does it in the right way, much better than we could do? In fact, we would seem to be morally culpable if we did not let the professional handle the nontrivial job rather than possibly botch it ourselves. Third, can we justify God letting the fawn suffer even a bit to give us a chance to rescue it? This would be an instance of God allowing the significant evil of suffering be a means to benefit us in our soul-building, all the while temporarily withholding the significant good from other living beings. It is not that God ought to wait a bit to give us a chance to rescue the fawn and put out the fire; God should have prevented the fire in the first place, since once started it affects the basic needs of many living beings.

Ultimately, God's temporary-delay solution violates the Pauline Principle of not doing evil—here to living beings—that good may come. If the evil resulting from the delay is significant, the Pauline Principle is violated; if the resulting evil is insignificant, it does not count against God's goodness and does not make for our character-building either.

Sterba's response ultimately becomes untenable when death is included among the significant natural evils. It is reasonable to include death, since life is a prerequisite for satisfying all needs and realizing all goods. Thus, to carry out Sterba's scenario, where God prevents significant natural evil to all living beings, sentient and non-sentient, satisfies their basic needs, and provides for their basic goods, when the satisfaction of basic needs does not contravene human basic needs, God would either have to give immortality to most living beings (depending on whether Sterba accepts eating meat and fish or root vegetables as a basic human need that would not be met without death) or exclude life from being a basic need. However, Sterba's examples treat life as a basic need. They involve either the evil of taking life itself (the fawn, Matthew Shepard) or fulfilling a basic need like freedom that presupposes that the being is alive.

In short, if to be good God is required by Sterba's Natural Evil Prevention Requirement IV to prevent significant natural evils for nonrational sentient living beings, and by Requirement VII for non-sentient living beings as well, then God would have to intervene to such an extent and in such a way that there would be no natural laws. God would be required to meticulously operate the world by divine intervention. Given the variety, "degree and amount" of natural evil in the world (Sterba 2019, p. 11), little regularity of causal relations would be left for us to calculate how to act. The result would be that, with God's intervention replacing natural causal relations, we would be unable to plan or act rationally, for all events would depend on God's actualization with the prevention of evil in mind. God alone would determine the most propitious outcomes.

To protect morally significant freedom and the human ability to plan and act rationally in the world, which is necessary for the greater good of having moral agents that do a significant amount of good, God will respect the natural laws that govern the world that he created. Moreover, if the universe operates by natural laws, and if living beings are natural beings, they will be affected by those laws, other natural beings, and natural events, sometimes to their benefit and sometimes not.[7]

## 13. Nomic Regularity

Sterba rejects this critique: "[T]here is no reason to think that God's (intervening to prevent natural evils) would adversely affect the nomic regularity and development from disorder to order of our world, leading to less good overall" (Sterba 2019, p. 169). Such a world would still have nomic regularities. As we noted above, he argues for constrained intervention.

However, one cannot have it both ways: on the one hand, that the degree and amount of significant natural evil are great enough to justify a claim that God does not exist, since a good God should be much more involved in intervening to prevent significant natural evil to sentient and non-sentient living beings alike. On the other hand, that divine miraculous intervention would not be great enough to disturb natural laws so that God's intervention would leave them and the rational deliberation and action they make possible to be fundamentally undisturbed.

Sterba responds that there could have been a different set of natural laws that did not result in significant natural evils (Sterba 2019, p. 63), but he provides neither a description of what such a world would be like nor an accounting of the degree and amount of evil that would result that would support his claim.

---

7 For a natural law theodicy, see (Reichenbach 1982, chp. 5).

### 14. The Threat of Deism

It might be suggested that, in replying to Sterba's atheological arguments, the advocate of a natural law theodicy promotes a position that likewise is unacceptable to many theists. Whereas atheists claim that God does not exist, deists claim that God, though existing, is absent from and uninvolved in the world. Because he is perfect, God created a perfect world, and once a perfect world existed, God has no reason to intervene. More importantly for our discussion, regular, frequent divine intervention would make human free action impossible, since no necessary or regular causal relations would hold between events to enable rational calculation and implementation of potential action.

On the contrary, however, a natural law theodicy need not be deist. What has been argued above is that to eliminate all significant evil, as Sterba suggests, a world *operated by* divine meticulous intervention would be necessary, and that such a world would be incompatible with agents knowing how to act and exercising morally significant freedom. However, a natural law theodicy does not eliminate divine intervention (Reichenbach 2016, pp. 225–29). Neither does divine intervention dispense with laws of nature; they operate before, during, and after the intervention. Rather, God intentionally introduces new features into the setting. As C.S. Lewis puts it, miracles are "an interference with nature by a supernatural power," an insertion of a new event into nature by a wise and powerful agent (Lewis 1960, p. 5).

As active beings, we frequently intervene in natural events in ways that interfere with the operation of natural laws. When I hold a rock, preventing it from falling, I interfere with the law of gravity. I do not violate the law of gravity; it still applies. However, I have introduced new events into the natural system that affect how the law of gravity functions in this case. Our limited intervention does not destroy our ability to recognize natural laws but presupposes that recognition. Similarly, the occasional divine intervention or miracle does not destroy our ability to recognize natural laws and is consistent with the character of a good God who does intervene (Lewis 1960, pp. 57–58). However, Sterba's requirements of eliminating all significant natural evil, given its "degree and amount," far exceed the presence of occasional divine interventions.

It might be objected that the theist cannot explain why God allows any particular evil, since preventing that one additional evil would not affect our ability to calculate rational action. If God can remove 100 evils, why not this one—101. Of course, the same objection can be repeated regarding evil 102, and so on, so that ultimately God would be obligated to remove all evils. However, to do so, we have argued, would mean that God would have to operate the world by meticulous divine interference, which would remove natural laws and the human ability to rationally calculate action. Since removing all evils is incompatible with the greater good of having free, rational moral agents, God must draw the line determining evils he can and does remove. From our perspective, and perhaps from God's, it would appear that that line is arbitrary, but the line must be drawn at some point (van Inwagen 2006, p. 105).[8]

### 15. Conclusions

Sterba thinks that he can resuscitate the atheologian's argument by appealing to the amount and degree of significant or horrendous moral and natural evils that would concern us and, particularly, the just state. Good beings and just states should intervene in human and natural events to prevent significant and horrendous evils as far as they easily can, without creating greater evil or losing significant freedom, rights, and basic goods, even if they have to restrict the freedom of some. Since God is all-powerful and all-knowing, he not only should but can easily eliminate these evils by intervening somewhere along

---

8   Our argument for God not eliminating all evil, based on the premise that it is good that moral agents exist and that having moral agents requires freedom of choice and action, differs from van Inwagen's, who bases it on the grounds that if he did so, God would frustrate his desire to reconcile all persons to himself (van Inwagen 2006, p. 88).

the causal chain. The degree and amount of these evils in the world shows, he claims, that God does not exist.

However, if we follow Moral Evil Prevention Requirement 1—the moral requirement that a good being must prevent all the significant moral evils when it can be easily done, without violating anyone's rights, and if we grant God omni-properties of power and knowledge, we conclude that God must eliminate all moral evil. Similarly, if we follow Natural Evil Prevention Requirements IV and VII, we conclude that God must eliminate all significant natural evil that he can do easily without infringing on human basic needs. We have contended that what constitutes significant and worst evils is a subjective, comparative concept, for what is significant or worst depends on what individual persons conceive to be significant, horrendous, or worst. Suppose that God prevents or removes all evils of level 7 magnitude. Instances of significant or worst evils would remain, namely evils of level 6 magnitude. According to Sterba's Requirements, God now would be obligated to remove or prevent all significant or worst evils, namely, those of level 6 magnitude. Were these removed, by the same argument, evils of level 5 magnitude would be significant and the worst and must be prevented or removed, and so on. Hence, Sterba's demand to mitigate all significant evil leads to the necessity of removing all evils. His scenario of constrained intervention is not a viable alternative, for either the evils are trivial and not significant enough to count against God's goodness, or else significant enough to require God's intervention, resulting in God operating the world by meticulous, divine intervention. However, this would remove our morally significant freedom to conceive or act, whether understood in a Plantingian or Sterban sense. Thus, there are good and sufficient reasons to doubt that Sterba has succeeded in defeating a freedom-based defense or theodicy.[9]

**Funding:** This research received no external funding.

**Conflicts of Interest:** The author declares no conflict of interest.

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
