# Peer review of "On James Sterba’s Refutation of Theistic Arguments to Justify Suffering"

_religions, doi:10.3390/rel12010064_

Round 1
Reviewer 1 Report
The author is no doubt correct that Sterba will not be cause to change his position by this essay. No doubt Sterba will consider that he has good answers to the objections raised. But the paper engages a broad swathe of Sterba's thinking in a way that can advance the discussion.
Author Response
There were no concrete suggestions to reply to in this reviewer's report.
Reviewer 2 Report
1. I have not read either of the pieces by James Sterba that the author criticizes. While this is not ideal for the purposes of reviewing the essay it does have at least one advantage: it puts me in a good position to discern how effectively the author conveys the substance of Sterba’s arguments. I find that it doesn’t do this very well. The 18-point reconstruction offered on lines 91-151 do not amount to a unified argumentative structure, and seem to combine quotes from Sterba with objections and responses, and with interpretive claims presented by the author. I cannot discern from this discussion how Sterba’s original argument is supposed to work. In particular, it is not clear to me why Sterba thinks that consideration of the duties of a just state is relevant for thinking about God; this seems potentially interesting, but comes across as unmotivated in this essay. In an essay of this kind one wants to get the sense that the arguments being criticized deserve serious consideration; I don’t get that sense here.
The plan of reconstructing Sterba’s argument in a dedicated section is a good one, but it needs to be executed more cleanly. In addition the relationship between material from Sterba’s texts presented later in the essay and the numbered exposition is not always clear.
2. The overall organization of the essay makes for difficult reading. The rationale for the ordering of topics is not clear. For example, the author leads off with an objection to Sterba’s conception of ‘significant freedoms’, but this takes place before we have Sterba’s actual argument on the table. Subsequent to the reconstruction of Sterba’s argument the author moves through a number of discrete criticisms, but again, the reason they are discussed in the order in which they appear is not clear.
In general, the structure of the essay needs to be tightened and clarified. I recommend that the author provide a roadmap at the beginning of the essay, listing the several topics in relation to Sterba’s arguments that they plan to discuss. It would be ideal if there were a natural ordering of these topics, such that each follows naturally on the preceding.
3. It is not obvious that any world in which God prevents the occurrence of significant evils will be a world entirely devoid of efficacious human action, as the author claims. There are many things that humans fail to do no matter how hard they try; but these limitations to the reach of human agency do not seem to make rational planning pointless. I do not see why a further constriction of the reach of human abilities— such as the failure of any attempts to bring about bad results— would be a kind of situation in which human agency has no point. It might also be worth mentioning middle knowledge in this connection; the author does not mention this important topic, and I cannot tell from the essay whether or not Sterba brings it into his arguments.
4. The theodicy presented by the author as a counter to Sterba is not very strong. I do not see any reason to think that any cosmos in which some set of natural laws obtain will be a cosmos containing natural evils. The author objects to Sterba’s noting this by saying that ‘we have no idea’ what natural laws these might be. But this is simply an argument from ignorance.
5. There is something of a core argument in the essay, one that has considerable virtues. This is the argument that the author cites in their abstract and conclusion; it involves objections to the thought that God should prevent more of ‘the worst’ evils than God apparently does. It is somewhat unfortunate that much of the material that comprises this core has been presented by other philosophers in the past. In particular I strongly recommend that the author familiarize themself with Peter Van Inwagen’s Gifford lectures, published in 2008 as The Problem of Evil (Oxford University Press). Van Inwagen presents a version of the same response, in the context of an approach to the problem of evil that I think the author will find sympathetic and useful.
6. I strongly recommend that the first sentence of the essay, which refers to Sterba’s own religious commitments, be deleted. An essay such as this should focus on arguments rather than on the (assumed or actual) religious commitments of the philosophers who advance them.
Author Response
- I have not read either of the pieces by James Sterba that the author criticizes. While this is not ideal for the purposes of reviewing the essay it does have at least one advantage: it puts me in a good position to discern how effectively the author conveys the substance of Sterba’s arguments. I find that it doesn’t do this very well. The 18-point reconstruction offered on lines 91-151 do not amount to a unified argumentative structure, and seem to combine quotes from Sterba with objections and responses, and with interpretive claims presented by the author. I cannot discern from this discussion how Sterba’s original argument is supposed to work. In particular, it is not clear to me why Sterba thinks that consideration of the duties of a just state is relevant for thinking about God; this seems potentially interesting, but comes across as unmotivated in this essay. In an essay of this kind one wants to get the sense that the arguments being criticized deserve serious consideration; I don’t get that sense here.
I begin the article by quoting one of Sterba’s own versions of his argument, along with his three Moral Evil Prevention Requirements. This should make clear exactly the argument advanced by Sterba that I want to address. I also tried to state clearly Sterba’s second argument from the Pauline Principle, isolating any of my clarification comments.
Regarding Sterba’s appeal to the just state, I suggest in a section entitled “Sterba’s Defense of Premise 2,” the motivation behind his comparison of a just God with a just state (they both invoke Sterba’s ethical principles) and note why I am not undertaking a criticism of that analogy.
The plan of reconstructing Sterba’s argument in a dedicated section is a good one, but it needs to be executed more cleanly. In addition the relationship between material from Sterba’s texts presented later in the essay and the numbered exposition is not always clear.
By employing more headings in the text that refer specifically to his argument, I connected my discussion and criticisms with specific premises of his original argument.
- The overall organization of the essay makes for difficult reading. The rationale for the ordering of topics is not clear. For example, the author leads off with an objection to Sterba’s conception of ‘significant freedoms’, but this takes place before we have Sterba’s actual argument on the table. Subsequent to the reconstruction of Sterba’s argument the author moves through a number of discrete criticisms, but again, the reason they are discussed in the order in which they appear is not clear.
In general, the structure of the essay needs to be tightened and clarified. I recommend that the author provide a roadmap at the beginning of the essay, listing the several topics in relation to Sterba’s arguments that they plan to discuss. It would be ideal if there were a natural ordering of these topics, such that each follows naturally on the preceding.
I provide the suggested roadmap of the essay in the opening of the article, put Sterba’s own arguments in the first section of the article, and significantly restructured the essay, especially in the first part, around the premises of Sterba’s original argument. The natural order—where my criticisms of his premises are followed by Sterba’s replies, and these in turn are considered—is followed and strengthened. Throughout the article, I added verbal guidelines to make sure that the structure is clear.
- It is not obvious that any world in which God prevents the occurrence of significant evils will be a world entirely devoid of efficacious human action, as the author claims. There are many things that humans fail to do no matter how hard they try; but these limitations to the reach of human agency do not seem to make rational planning pointless. I do not see why a further constriction of the reach of human abilities— such as the failure of any attempts to bring about bad results— would be a kind of situation in which human agency has no point. It might also be worth mentioning middle knowledge in this connection; the author does not mention this important topic, and I cannot tell from the essay whether or not Sterba brings it into his arguments.
Sterba’s argument is based on the contention that the degree and amount of evil extant in the world cannot be justified. I have replied that if there is so much significant moral and natural evil, then God, if God existed, would have to operate the world by meticulous intervention to address it. If, on the other hand, there is not much significant evil, or if the evils are trivial and reparable, such that restrictions of human knowledge and freedom are not seriously affected, then his argument loses much of its power, for they do not require divine intervention. They are, as his illustration suggests, like stepping on someone’s foot exiting the subway.
I did not address middle knowledge. Sterba himself believes that his argument circumvents this discussion, and so I did not feel that I had to address it. It would take the discussion off in an entirely different direction.
- The theodicy presented by the author as a counter to Sterba is not very strong. I do not see any reason to think that any cosmos in which some set of natural laws obtain will be a cosmos containing natural evils. The author objects to Sterba’s noting this by saying that ‘we have no idea’ what natural laws these might be. But this is simply an argument from ignorance.
I have mostly dropped my development of a natural law theodicy, only considering what pieces Sterba objects to. But I do argue that if sentient and nonsentient living beings are natural beings, they will be affected by natural laws and natural events, sometime propitiously and sometime not. I would love to see the reviewer’s development of a natural law system that would have no loss, negative, or limiting effects on sentient, natural beings (no natural evil).
I have altered the alleged argument from ignorance and put the burden of proof on Sterba to show what his world of altered natural laws would be like and how it would reduce the amount and severity of natural evil. He does not argue that natural evil would be eliminated.
- There is something of a core argument in the essay, one that has considerable virtues. This is the argument that the author cites in their abstract and conclusion; it involves objections to the thought that God should prevent more of ‘the worst’ evils than God apparently does. It is somewhat unfortunate that much of the material that comprises this core has been presented by other philosophers in the past. In particular I strongly recommend that the author familiarize themself with Peter Van Inwagen’s Gifford lectures, published in 2008 as The Problem of Evil(Oxford University Press). Van Inwagen presents a version of the same response, in the context of an approach to the problem of evil that I think the author will find sympathetic and useful.
In the final section, I have briefly noted van Inwagen’s presentation and in footnote 8 noted how it differs from my argument.
- I strongly recommend that the first sentence of the essay, which refers to Sterba’s own religious commitments, be deleted. An essay such as this should focus on arguments rather than on the (assumed or actual) religious commitments of the philosophers who advance them.
I completely rewrote the first paragraph.
Is the article adequately referenced? I have provided extensive references to Sterba’s work in the text, to show that my references to his argument and replies are accurate.
Reviewer 3 Report
The author shows a refreshing dose of common sense in his rigorous analysis of the ambiguities in Sterba's work. Sterba's contention that God is morally obligated to intervene to prevent only "horrendous evils" is reduced to a level of absurdity by the author when he backs Sterba's position into a logical corner and points out that "evils" depend on personal perspectives. At which level of evil should God not intervene? I think this is one of the more salient points the author makes in his article.
The complex nature of suffering for both sentient and non-sentient beings is effectively dealt with by the author with his references to concrete examples of scenarios involving both. This is especially true with the author's reference to his experience at the slave castles in Ghana. What the slaves experienced was not trivial; but Sterba's response does indeed trivialize their suffering. I think it was Whitehead who said something to the effect that the major sin of philosophers is over-generalizations. The author highlights this "sin" of Sterba's with his (the author's) use of concrete examples that appeal to humane common sense.
The style and composition of the article makes it very readable. The author is consistently rigorous in his analysis of Sterba's work without being caustic or acerbic in tone. It seems to me that he is fair and balanced in presenting Sterba's points of view, while consistently challenging Sterba's positions.
The author could have taken to task Sterba more at length on Sterba's use of the just state analogy. First, Sterba's characterization of a just state is ambiguous, as the author briefly points out. Second, Sterba assumes to know what an omniscient, omnipotent, omnipresent, benevolent Creator would want in a just state. The analogy breaks down between a just state as created by humans and a just state that God would create. Sterba assumes he knows the mind of God.
This last point also applies to Sterba's contention that God should intervene to prevent "horrendous evils." If one is "attacking" God based on his omni-properties , then one should also realize that these omni-properties are beyond the knowledge and understanding of humans. So Sterba ends up assuming, for the sake of argument, that God has these properties, while ignoring that the nature of these properties are beyond our grasp. I would suggest that the author come down a little bit harder on Sterba's application and non-application of God's properties.
This is a well-written and well-argued article. It should interest anyone, who may be interested in traditional theodicy. I think his critique of such a well-known and popular philosopher as James Sterba also will interest readers.
At the end of the article, however, I was left with the notion that, well, the author has "made the world safe for free-will" again. This is done by appeal to natural law; and the occasional miracle makes the world safe for God's intervention.
Without the occasional miracle, however, I am left with a deistic view of the world and God. I don't mind this at all; but it may be something to point out to the author. I realize that the author is attempting to refute Sterba's refutation of the theistic arguments to justify suffering and he may not be trying to save free-will and a traditional conception of God; but some readers may conclude that this part of his article needs a little more attention.
Author Response
I added a section at the end of the article, showing how my natural law position differs from and does not invoke deism.
Round 2
Reviewer 2 Report
This draft is much improved. At present I have only one suggestion, which is that the author acknowledge that the argument they present on p. 7 ("If God eliminates highest evils of level 7, then the question arises why a good, omniscient, and almighty God is not causally involved in the world to remove evils of level 6, since these are now the most significant or serious evils") is also presented by Peter Van Inwagen (2006).